# Rolling Motion of a Soft Microsnowman under Rotating Magnetic Field

**DOI:** 10.3390/mi13071005

**Published:** 2022-06-26

**Authors:** Gokhan Kararsiz, Yasin Cagatay Duygu, Louis William Rogowski, Anuruddha Bhattacharjee, Min Jun Kim

**Affiliations:** 1Department of Mechanical Engineering, Southern Methodist University, Dallas, TX 75275, USA; gkararsiz@smu.edu (G.K.); aduygu@smu.edu (Y.C.D.); abhattacharjee@smu.edu (A.B.); 2Applied Research Associates, Inc. (ARA), 4300 San Mateo Blvd. NE, Suite A-220, Albuquerque, NM 87110, USA; lrogowski@ara.com

**Keywords:** swarm control, alginate microbot, magnetic control

## Abstract

This paper demonstrates a manipulation of snowman-shaped soft microrobots under a uniform rotating magnetic field. Each microsnowman robot consists of two biocompatible alginate microspheres with embedded magnetic nanoparticles. The soft microsnowmen were fabricated using a microfluidic device by following a centrifuge-based microfluidic droplet method. Under a uniform rotating magnetic field, the microsnowmen were rolled on the substrate surface, and the velocity response for increasing magnetic field frequencies was analyzed. Then, a microsnowman was rolled to follow different paths, which demonstrated directional controllability of the microrobot. Moreover, swarms of microsnowmen and single alginate microrobots were manipulated under the rotating magnetic field, and their velocity responses were analyzed for comparison.

## 1. Introduction

Recent advances in robotics field endeavor to create viable deformable robots in macro and micro scales. Soft robotics is a field of research that has opened new prospects to solve the challenges of authentic tasks that rigid robots cannot overcome [1,2,3,4,5]. Higher degrees of freedom and small stiffness values of soft robots make them a better candidate to achieve high mobility tasks thanks to their multimodal locomotion [6,7,8]. Their small stiffness also makes them characteristically adaptive, which can enable manipulation on rough environments [9,10,11,12].

Soft microrobotics has been attracting researchers’ interest due to its advantages, such as biocompatibility, flexibility, adaptability, and motility [13,14]. A lot of work has been devoted to develop soft microrobots for drug delivery [15], cell delivery [16], microsurgery applications [17], and microdevice imaging [18,19]. In order to manipulate soft microrobots, magnetic [20,21], chemical [22], acoustic [23], light [24], hydrolic [25], and pneumatic [26] sources have been employed. Among these methods, magnetic field is widely used because of its high performance in complex manipulations wirelessly [27]. In this work, we use a uniform rotating magnetic field for microrobot manipulation. The rotating magnetic field enables creating magnetic torques, which allows rolling motion of the soft alginate microrobots on the surface without slippage. Several fabrication methods have been reported in the litearture, including 3D printing [28], litrography [29], and microfluidic [30] methods. It is reported in [31] that all fabrication methods mentioned above are complex and costly. We chose the centrifuge-based microfluidic droplet method fabrication because of its success in encapsulation process with using inexpensive and widely available equipments.

In previous work, a microsnowman composed of two rigid microparticles (10 micron diameters) was investigated for swimming propulsion within synthetic mucus [32]. These microsnowmen were non-deformable and relied on specialized fluid properties to propel far from the surface [33]. Here, we investigate a deformable microsnowman composed of alginate-based hydrogels and propel them along the surface of a substrate using rolling motion.

The rest of the paper is organized as follows. In Section 2, the background information regarding the fabrication of the microsnowman robot, experimental system, and the control strategies for magnetic field are discussed. In Section 3, the motion analysis for microsnowman, tracking results for various paths, and swarm control results are illustrated. In Section 4, a conclusion is presented at the end.

## 2. Materials and Methods

### 2.1. Fabrication

Fabrication process of the alginated-based soft microsnowman is shown in Figure 1. Alginate particles around 100 μm in size are fabricated by a centrifuge-based microfluidic droplet method. First, a 5% concentration sodium-alginate (Sigma Aldrich A1152-500 G, St. Louis, MO, USA) solution is mixed with 5% ${\mathrm{Fe}}_{2}O_{3}$ (Sigma Aldrich 647106-100 G, MO, USA) particles to impart the magnetic property on the artificial cell. Then, a 1.5 Ml centrifuge tube (Eppendorf T9661-500EA, Enfield, CT, USA) is drilled from the center of the cap, and calcium chloride (${\mathrm{CaCl}}_{2}$) solution (5% v/w, Sigma Aldrich C1016-500 G, St. Louis, MO, USA) is placed with a 1.5 mm gap between the needle tip and the calcium chloride solution. The needle (BD precisionglide${}^{TM}$ REF305109 27 G X 1/2 inch, Franklin Lakes, NJ, USA) is pushed from that hole. A tube with alginate solution is centrifuged with 1000 relative centrigual force (rcf) for 1 min. An Eppendorf microcentrifuge 5446 is used for this experiment. The microsnowman or single alginate micro-droplets are randomly generated by the centrifugal force at the tip of the needle according to their eject velocity. When the sodium-alginate droplets contact with the calcium chloride solution, the alginate solution is crosslinked with iron-oxide particles encapsulated [31]. In this paper, we refer two aggregated alginate particles as the soft microsnowman. After encapsulation process, a magnetic field generated by a permanent magnet was used to align the iron(III) oxide particles inside of the soft alginate particles for 10 min.

### 2.2. Experimental Methodology

The hardware setup for experiments is shown in Figure 2a. The fabricated soft alginate particles were put into a PDMS chamber (15 mm diameter) with supported micro cover glass (No 1, 22 × 30 mm). Then, the chamber was placed in the middle of an approximate triaxial Helmholtz coil system. The detailed information regarding the coil system can be found in [34]. In order to visualize the soft microrobots in motion, 4× plan Leica objective lens was used with Leica DM IRB microscope. The experimental videos were captured by a CMOS camera (PointGrey FL3-U3-13Y3M-C, Wilsonville, OR, USA) with resolution 512 pixels × 640 pixels (2.6925 μm/pixels) at 30 frame per second (fps). A digital signal is created by a LabVIEW interface to control the magnetic field inside the coil system. This signal is converted to analog signal in DAQ device (National Instruments BNC-2110, Austin, TX, USA). The converted analog signals are sent into three identical power supplies (KEPCO BOP-20-5M, Fort Lee, NJ, USA) to generate uniform magnetic field in each three axes.

Exploiting the magnetic properties of iron(III) oxide particles, the soft microsnowman robots are driven wirelessly by manipulating the magnetic field. In this paper, the translation of the robots was achieved by rolling locomotion. The effect of rolling motion under the rotating magnetic field is shown in Figure 2b. A 90-degree angle exists between the rotation axis and the propulsion axis of the microsnowman.

The applied rotating magnetic field for rolling motion can be written as
(1)$$\begin{array}{c} B=\left[\begin{array}{c} B\sin\theta \cos\omega t\\ B\cos\theta \cos\omega t\\ B\sin\omega t \end{array}\right] \end{array}$$
with the direction vector
(2)$$\begin{array}{c} \hat{n}=\left[\begin{array}{ccc} -\cos\theta & \sin\theta & 0 \end{array}\right] \end{array}$$
where the coefficients *B*, θ, ω, *t* stand for the amplitude, the heading angle, frequency, and time in seconds for the rotating magnetic field, respectively.

### 2.3. Kinematic Model of Microsnowman Robot

The kinematics for the rolling soft microsnowman is inspired from 2D unicycle robot dynamics. This model was used previously in [35] for rolling motion. The kinematic model for the rolling soft microsnowman is described by
(3)$$\begin{array}{cc} \dot{x}\left(t\right) & =v(t)\cos\theta (t) \end{array}$$
(4)$$\begin{array}{cc} \dot{y}\left(t\right) & =v(t)\sin\theta (t) \end{array}$$
(5)$$\begin{array}{cc} \dot{\theta }\left(t\right) & =u(t)\;\;\; \end{array}$$
where the x(t) and y(t) are the location in Cartesian space, v(t) represents the velocity of the robot. The turning rate of the rotational magnetic field is u(t). The symbol θ(t) shows the heading angle, which can be seen in Figure 2b. The open-loop control input only affects the heading angle of the robot during its manipulation. The magnitude of the magnetic field is set to a fixed value during the experiments, which results in constant velocity.

## 3. Results

Various experiments were performed to understand the motion of the soft microsnowman under rotating magnetic fields. For each experiment, a different microsnowman robot was utilized. Due to geometrical differences, please note that each robot has its own characteristics in terms of rolling performance. The microsnowman robots were tested with the approximate Helmoltz coil system inside a PDMS chamber filled with DI water.

### 3.1. Velocity Analysis

The velocity analysis was performed to understand the controllability of the microsnowman robots at various frequencies (1 Hz, 3 Hz, 5 Hz, 7 Hz). The results can be seen in Figure 3. The experiment was carried out with four different soft microsnowmen. The figure shows that the microsnowman experience a near linear increase in velocity as the rotational frequency is increased. To show the linearity of the results, a linear fitting is employed to the average velocity values of the four microsnowmen. Statistical analysis and graph for the plotted linear fit are given by Table 1 and Figure 3, respectively. The velocity data was obtained from a custom MATLAB script by applying post-processing methods. The location information was extracted from the centroid position for each frame. By using image binarization with grayscale thresholding, the centroid position is tracked with *x* and *y* coordinates for each frame. Then, the velocity data is obtained from the location data.

### 3.2. Directional Controllability

The trajectory of a microsnowman robot following the letters ‘S’, ‘M’, and ‘U’ with the velocity vs. time graphs is shown in Figure 4. The microsnowman was rolled on the substrate surface under a uniform rotating magnetic field during the experiments. The magnitude and frequency of applied magnetic field in clockwise direction were set to 8 mT and 3 Hz for the letter ‘S’ and, 5 mT and 1 Hz for the letters ‘M’ and, ‘U’, respectively. The open-loop control was utilized with the heading angles 0, 90, 135, 180, 225, and 270 degrees. During the experiment, pre-defined corner locations was selected by user as an goal point. After the robot was reached to the corner point, the heading angle was changed according to the pattern. Figure 4 shows that distortions in the trajectory increased from the letter ‘S’ to ‘U’. This result stems from the deformable body of the microsnowman robots. After translation on the substrate surface over time, the spherical shape was distorted and created a deviated path.

### 3.3. Swarm Control

By exploiting the global magnetic input, a swarm consisted of single and soft microsnowman alginate microrobots were controlled. In Figure 5a–d, two microsnowmen and two single alginate particles followed a rectangular path. It is seen that the traveled distances with the microsnowman robots improved significantly in comparison to single particles. The velocity data is shared in Table 2. Additionally, the strech-like behaviour was observed in the trajectory of Microsnowman 1 due to distorted alignment (see Appendix A). The reason for this behavior is that the soft double alginate particle tried to align itself to a new dipole orientation.

## 4. Conclusions

In summary, this paper presents an alginate-based microsnowman soft robots for rolling on the substrate surface under a uniform rotating magnetic field. A centrifuged-based microfluidic droplet method was utilized to create snowman-shape robots in microscale using alginate-based hydrogel. To characterize the motion capacity, the velocity responses of the rolling microsnowman robot were investigated at incremental frequencies. By using swarm of microsnowmen and single alginate particles, the performance of the robots were compared. As a future work, more complex locomotion modes, such as, tumbling and pivot walking, will be implemented, and we will try to fabricate three or more aggregated alginate particles with the pressure-based droplet method.

## Figures and Tables

**Figure 1 micromachines-13-01005-f001:**
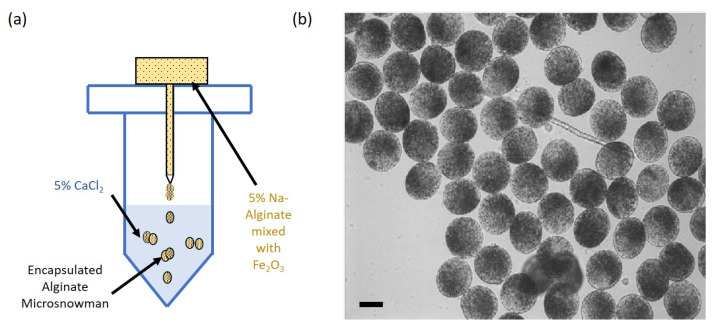
Fabrication of soft alginate particles by a centrifuge-based microfluidic droplet method. (**a**) A droplet microfluidic device was used for the fabrication of the alginate-based microsnowman with encapsulated iron-oxide nano particles. (**b**) The bright field image of the fabricated alginate particles. Single and microsnowman type robots can be seen from the image. The scale bar is set to 100 μm.

**Figure 2 micromachines-13-01005-f002:**
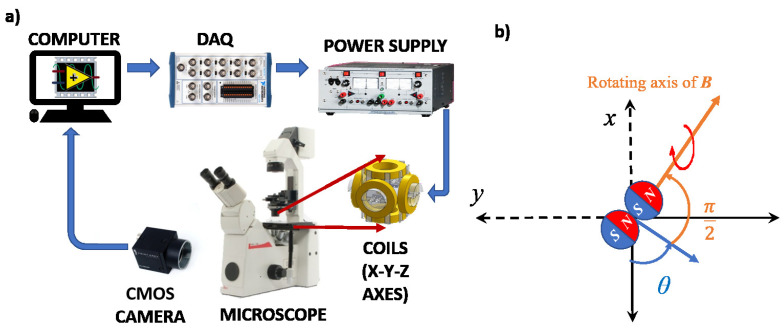
(**a**) The experimental hardware scheme is shown. The videos and the images are captured by a CMOS camera attached to the microscope. (**b**) The rolling motion on Cartesian space is depicted. The symbol θ in blue represents the current heading angle.

**Figure 3 micromachines-13-01005-f003:**
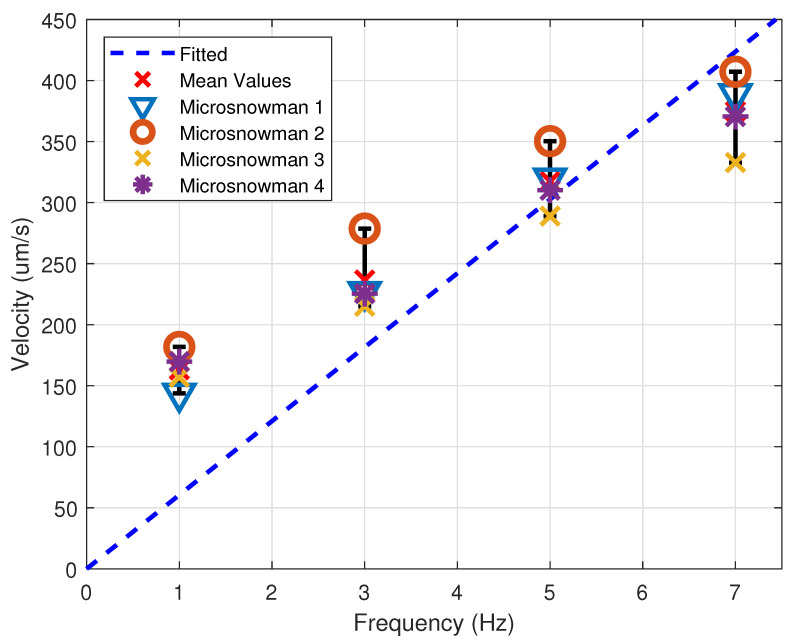
Velocity vs. Frequency plot for four different soft microsnowmen with a linear fit for mean velocity. Error bars are represented with black color.

**Figure 4 micromachines-13-01005-f004:**
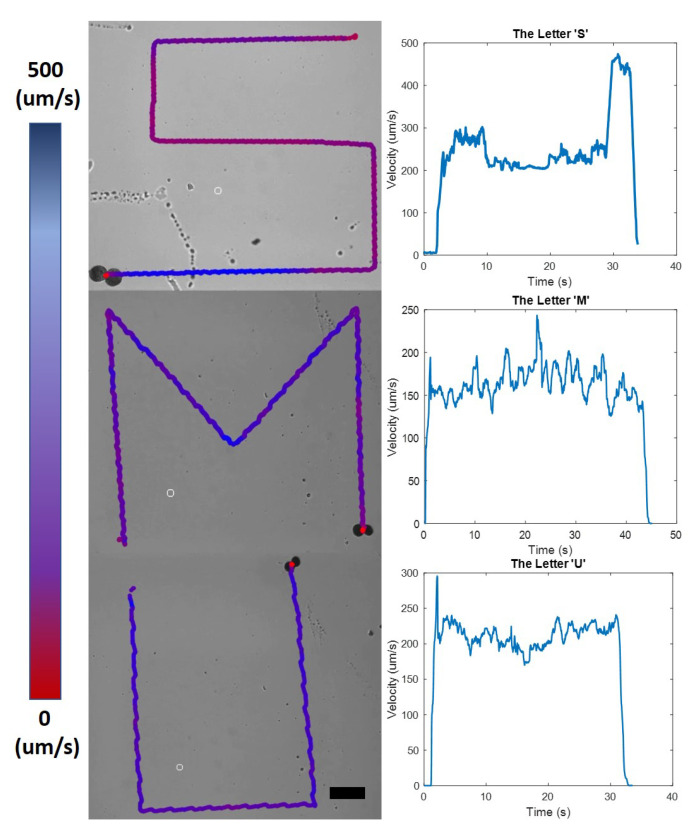
Trajectory of a microsnowman robot using open-loop control; the microrobot was rolled to follow the patterns ‘S’, ‘M’, and ‘U’. For each figure, the robot was located at the finish position. The gradient colorbar shows the velocity of the microsnwoman robot in each location. The scale bar is set to 400 micron. The travel time is 33, 45, and 33 s for the letters ‘S’, ‘M’ and, ‘U’, respectively.

**Figure 5 micromachines-13-01005-f005:**
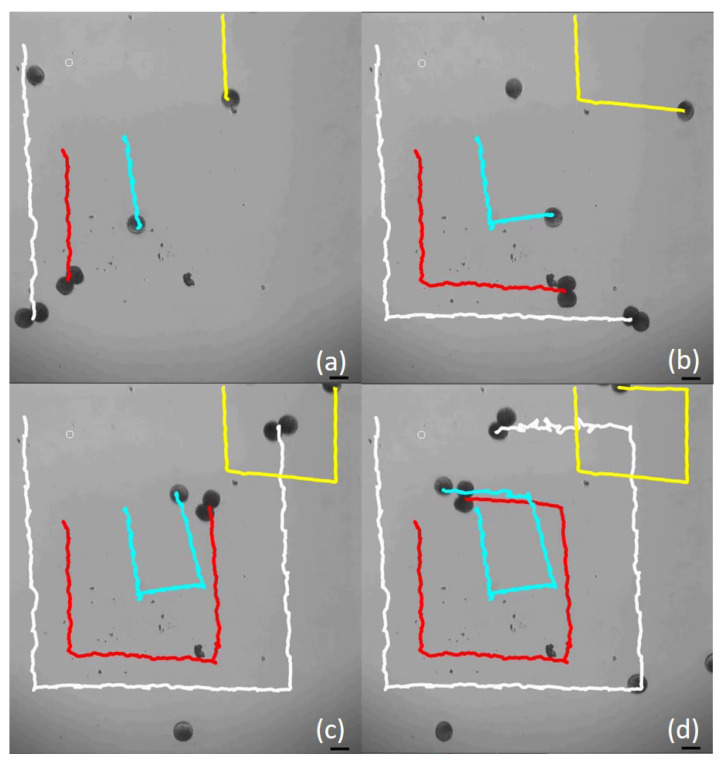
Swarm control of soft microsnowmen and single alginate microrobots. In frames from (**a**–**d**), the translation distances are shown for a rectangular path. The scale bar is set to 100 micron.

**Table 1 micromachines-13-01005-t001:** Values for the curve fitting line in Figure 3.

Frequency	Mean Velocity	Standart Deviation	Coefficient of Variations
(Hz)	(μm/s)	(μm/s)	
1	163.01	14.22	202.15
3	236.56	24.76	613.09
5	317.37	22.09	487.76
7	374.89	27.56	759.67

**Table 2 micromachines-13-01005-t002:** Calculated Mean Velocities for Swarm Motion.

Microsnowman 1	Microsnowman 2	Single Particle 1	Single Particle 2
(White)	(Red)	(Turquoise)	(Yellow)
279.66 (μm/s)	180.47 (μm/s)	121.92 (μm/s)	143.39 (μm/s)

## Data Availability

All data for this study have been experimentally generated and have been included in this paper.

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
