# Peer review of "Rolling Motion of a Soft Microsnowman under Rotating Magnetic Field"

_micromachines, 2022, doi:10.3390/mi13071005_

Round 1

Reviewer 1 Report

This work provides a method using a rotating magnetic field for the manipulation of snowman-shaped soft microrobots, followed by the analysis of the directional controllability. The comparison of the velocities between the swarms of microsnowmen and a single alginate microrobot shows that the volume of the microsnowman robots significantly affect the velocity response, which is quite interesting. Overall, this work is attractive. Here are some questions related to the manuscript before it is accepted for publication.

1. As shown in Fig. 3, it seems that the green dotted line matches better than the proposed blue dotted line. The author shall explain the reason and give solutions for a better result.

2. In Section 3.2 Directional Controllability, the two experiments both show the controllability of the microsnowman, which is redundant -- Fig. 5 is enough for the understanding of the controllability. Fig. 4 is recommended to be replaced by the figures of velocity analysis, which directly shows the velocity response of the microsnowman under different frequencies. The colour can be used for representing the velocity, instead of the time sequence.

Author Response

Response to Reviewer 1 Comments

  1. As shown in Fig. 3, it seems that the green dotted line matches better than the proposed blue dotted line. The author shall explain the reason and give solutions for a better result.

If the frequency range is restricted around 1 to 7 hertz, the fit becomes in the figure below. The reason of this behavior is that there is a highly non-linear region around 0-1 Hz. Due to the great increase on the velocity from 0 to 1 Hz, the linear fit is distorted.  Please see the figure below for the linear fit with restricted frequency range. According to given fit below, the microsnowman robot supposed to have a velocity around 100 um/s at 0 Hz.

  1. In Section 3.2 Directional Controllability, the two experiments both show the controllability of the microsnowman, which is redundant -- Fig. 5 is enough for the understanding of the controllability. Fig. 4 is recommended to be replaced by the figures of velocity analysis, which directly shows the velocity response of the microsnowman under different frequencies. The colour can be used for representing the velocity, instead of the time sequence.

In the manuscript, the figure 4 is redacted and the velocity information is added to Figure 5. The colorbar is set in between 0 to 500 micrometer / second.

Reviewer 2 Report

     In the papers, the authors present alginate-based microsnowman soft robots for rolling on the substrate surface under a uniform rotating magnetic field. Their device has high repeatability. Some improvements should be done. 

1. In the introduction, they should mention some power sources for soft robots, like hydraulic or pneumatic, or others. 

Underwater Crawling Robot With Hydraulic Soft Actuators. Frontiers in Robotics and AI, 263.Eccentric actuator driven by stacked electrohydrodynamic pumps. Journal of Zhejiang University-SCIENCE A, 2022, 23.4: 329-334.

2. In figure.3, why the authors do not compare their theoretical and experimental results?

3. In fiugre.4 and figure.5, can their robot move at another angle, not like 90 degrees. 

4. Figure.6c and figure.6d, the bottom particles can not move, why does this happen?

5. Their video is interesting. Can they control the robots independently ? like two particles collide. 

Author Response

Response to Reviewer 2 Comments

  1. In the introduction, they should mention some power sources for soft robots, like hydraulic or pneumatic, or others. 

Underwater Crawling Robot With Hydraulic Soft Actuators. Frontiers in Robotics and AI, 263.Eccentric actuator driven by stacked electrohydrodynamic pumps. Journal of Zhejiang University-SCIENCE A, 2022, 23.4: 329-334.

The part reviewer mentioned was added to the introduction section of the manuscript. The changes are highlighted with orange color.

  1. In figure.3, why the authors do not compare their theoretical and experimental results?

In this manuscript, we only focused on the experimental work. Our aim is to extend this study with different actuation methods such as pivot walking or tumbling. We will definitely add the simulation results to the new manuscript.

  1. In fiugre.4 and figure.5, can their robot move at another angle, not like 90 degrees. 

If you check the Figure 5b for the letter M, the angle was set to 135 and 225 degrees, respectively to follow the path. Considering the limited workspace and high velocity of the particles, it is not easy to get data with different degrees by using open-loop control.

  1. 6c and figure.6d, the bottom particles can not move, why does this happen?

By using the global single magnetic input, all robots were moving in tandem. Due to limited workspace, some robots were not in the field of view at the beginning (Figure 6a and b). After the manipulation and following the path, the additional robots were appearing on the field of view (Figure 6c and d).

  1. Their video is interesting. Can they control the robots independently ? like two particles collide. 

Swarm control is achieved by exploiting the global input. Right now, all the robots are aligned to the direction of the magnetic field. However, their response to the magnetic field vary due to the heterogeneity of the robots. With the current setup by using wall constraints with uniform robots or preprogrammed path with heterogenous robots, a collision of the robots can be achievable.

Round 2

Reviewer 2 Report

The authors answered the questions carefully. 

In the introduction, it is not hydrolic, 

it is hydraulic.